# Association Study of Serotonin 1A Receptor Gene, Personality, and Anxiety in Women with Alcohol Use Disorder

**DOI:** 10.3390/ijms25126563

**Published:** 2024-06-14

**Authors:** Agnieszka Boroń, Aleksandra Suchanecka, Krzysztof Chmielowiec, Jolanta Chmielowiec, Milena Lachowicz, Aleksandra Strońska-Pluta, Grzegorz Trybek, Tomasz Wach, Pablo José González Domenech, Anna Grzywacz

**Affiliations:** 1Department of Clinical and Molecular Biochemistry, Pomeranian Medical University in Szczecin, Powstańców Wielkopolskich 72 Str., 70-111 Szczecin, Poland; agnieszka.boron@pum.edu.pl; 2Independent Laboratory of Behavioral Genetics and Epigenetics, Pomeranian Medical University in Szczecin, Powstańców Wielkopolskich 72 Str., 70-111 Szczecin, Poland; aleksandra.suchanecka@pum.edu.pl (A.S.); aleksandra.stronska@pum.edu.pl (A.S.-P.); 3Department of Hygiene and Epidemiology, Collegium Medicum, University of Zielona Góra, 28 Zyty Str., 65-046 Zielona Góra, Poland; chmiele@vp.pl (K.C.); chmiele1@o2.pl (J.C.); 4Department of Psychology, Gdansk University of Physical Education and Sport, 80-336 Gdansk, Poland; milena.lachowicz@awf.gda.pl; 5Department of Oral Surgery, Pomeranian Medical University in Szczecin, 70-111 Szczecin, Poland; g.trybek@gmail.com; 6Maxillofacial Surgery Clinic, 4th Military Clinical Hospital in Wroclaw, ul. Rudolfa Weigla 5, 50-981 Wroclaw, Poland; 7Department of Maxillofacial Surgery, Medical University of Lodz, 113 Żeromskiego Str., 90-549 Lodz, Poland; tomasz.wach@umed.lodz.pl; 8Facultad de Medicina, University of Granada, Avenida de la Investigación nº 11, 18071 Granada, Spain; pgdomenech@ugr.es

**Keywords:** alcohol use disorder, serotonin receptor 1A, personality, anxiety, rs6295

## Abstract

Alcohol use disorder is considered a chronic and relapsing disorder affecting the central nervous system. The serotonergic system, mainly through its influence on the mesolimbic dopaminergic reward system, has been postulated to play a pivotal role in the underlying mechanism of alcohol dependence. The study aims to analyse the association of the rs6295 polymorphism of the *5HTR1A* gene in women with alcohol use disorder and the association of personality traits with the development of alcohol dependence, as well as the interaction of the rs6295, personality traits, and anxiety with alcohol dependence in women. The study group consisted of 213 female volunteers: 101 with alcohol use disorder and 112 controls. NEO Five-Factor and State-Trait Anxiety Inventories were applied for psychometric testing. Genotyping of rs6295 was performed by real-time PCR. We did not observe significant differences in *5HTR1A* rs6295 genotypes (*p* = 0.2709) or allele distribution (*p* = 0.4513). The AUD subjects scored higher on the anxiety trait (*p* < 0.0001) and anxiety state (*p* < 0.0001) scales, as well as on the neuroticism (*p* < 0.0001) and openness (*p* = 0134) scales. Significantly lower scores were obtained by the AUD subjects on the extraversion (*p* < 0.0001), agreeability (*p* < 0.0001), and conscientiousness (*p* < 0.0001) scales. Additionally, we observed a significant effect of *5HTR1A* rs6295 genotype interaction and alcohol dependency, or lack thereof, on the openness scale (*p* = 0.0016). In summary, this study offers a comprehensive overview of alcohol dependence among women. It offers valuable insights into this complex topic, contributing to a more nuanced understanding of substance use among this specific demographic. Additionally, these findings may have implications for developing prevention and intervention strategies tailored to individual genetic and, most importantly, personality and anxiety differences.

## 1. Introduction

Alcohol use disorder (AUD) is considered a chronic, relapsing disorder characterised by compulsive alcohol seeking and drinking, the inability to regulate alcohol consumption, and experiencing negative emotional states in the absence of alcohol [1,2,3]. The prevalence of alcohol abuse in the community is considerable, with a high incidence of co-occurrence with emotional and mood disorders, including anxiety and depression [4]. Additionally, a significant contributor to craving and relapse is increased anxiety following alcohol withdrawal [5]. Genetics have been demonstrated to be a significant factor in the development of alcohol dependence, as evidenced by studies involving families, twins, and adoptions [6,7,8].

The serotonergic system, mainly through its influence on the mesolimbic dopaminergic reward system, has been postulated to play a pivotal role in the underlying mechanism of alcohol dependence [9]. In addition, personality traits that increase susceptibility to alcohol dependence—particular subtypes of alcohol dependence—have been linked to serotonin-related impulsivity and suicidal behaviour [10,11].

The serotonin (5-hydroxytryptamine, 5-HT) system is widely associated with regulating feelings, arousal, mood, impulsiveness, and reward. It is postulated that dysregulation of serotonin homeostasis is implicated in the pathogenesis of anxiety and depression. Functional changes in limbic serotonergic signalling have been reported following acute and chronic alcohol exposure [12,13]. Excessive 5-HT signalling may be a contributing factor in the aetiology of AUDs. It can induce cravings in abstinent patients [14,15]. Polymorphisms in genes encoding the serotonergic system either predispose to the development of AUD or are associated with specific AUD-related behaviours [16,17,18].

There are several ways in which alcohol affects serotonergic synaptic transmission in the brain. Several aspects of serotonergic synaptic function are altered by a single (i.e., acute) exposure to alcohol. For example, in humans, urinary and blood levels of serotonin metabolites are increased after a single alcohol consumption, which indicates increased 5-HT release [9]. Increased signalling at serotonergic synapses may be responsible for this increase. Animal studies have also shown that acute alcohol exposure increases serotonin levels in the brain [19,20]. This indicates that either there is an increase in the release of serotonin from serotonergic axons or that it is removed from the synapse at a slower rate. In areas of the brain that control the use of multiple substances, increased serotonin release has been observed after acute alcohol exposure [20].

The *HTR1A* gene is located at 5q11.2-q13. It encodes the G-protein-coupled serotonergic 1A receptor that binds to the endogenous 5-hydroxytryptamine. It is a Gi/Go-coupled receptor, and it mediates inhibitory neurotransmission. Serotonergic 1A receptors are abundant in the central nervous system (hippocampus, amygdala, cortex, nucleus accumbens, and septum). It has been shown that activation of the 5-HT1A receptor enhances dopamine release in the medial prefrontal cortex, hippocampus, and striatum. Additionally, it impairs cognitive function, learning, and memory by inhibiting glutamate and acetylcholine release in various brain regions, increasing impulsivity, and inhibiting addictive behaviour. The development of alcohol dependence is, therefore, likely to be associated with this activation [21].

Transcriptional regulation studies of the *HTR1A* gene have identified a putative functional GC single nucleotide polymorphism (SNP), rs6295, at 1009 bp upstream of the translation start site [22]. The G allele was initially found to be associated with committing suicide [23] and a reduced response to antidepressants [24]. In addition, this SNP has also been linked to several other disorders, including substance abuse, panic disorder [25], premenstrual dysphoric disorder [26], schizophrenia [27], and eating disorders [27]. The findings suggest that rs6295 might influence intermediate phenotypic traits shared by many neuropsychiatric disorders [28].

rs6295 may have a functional effect on the transcription of *HTR1A* by altering the binding of transcription factors, according to in vitro studies [22]. Specifically, the G allele does not bind to the NUDR/Deaf1 transcription factor. This results in increased rates of *HTR1A* transcription in raphe-derived neurons where NUDR/Deaf1 functions as a repressor but decreased rates in forebrain-derived neurons where it functions as a transcriptional activator [29,30]. This bifaceted effect has never been verified in human tissues. Furthermore, rs6295 affects the binding of the Hes1 and Hes5 factors, mainly expressed during neuronal differentiation [23,29,31]. Therefore, the rs6295 effect on early developmental transcription may differ from adult effects.

As alcohol dependence is influenced by many factors, including sex, genetics, and personality, our study aims to analyse the association of the rs6295 polymorphism of the *5HTR1A* gene in women with alcohol use disorder and the association of personality traits with the development of alcohol dependence, as well as a multivariate analysis, i.e., analysis of the interaction of the rs6295 polymorphism, personality traits, and anxiety with the alcohol dependence in women.

## 2. Results

The frequency distribution of genotypes and alleles of the *5HTR1A* gene rs6295 polymorphism accorded with the HWE in the AUD and control subjects (Table 1).

We did not observe significant differences in *5HTR1A* rs6295 genotypes (G/C 0.52 vs. G/C 0.42; G/G 0.28 vs. G/G 0.37; C/C 0.20 vs. C/C 0.21, χ^2^ = 2.612, *p* = 0.2709) nor allele distribution (G 0.54 vs. G 0.58; C 0.46 vs. C 0.42, χ^2^ = 0.570, *p* = 0.4513) when comparing the AUD group to the controls. Results are presented in Table 2.

We observed several significant differences in the mean scores of the analysed STAI and NEO-FFI traits when compared between groups. The AUD subjects scored significantly higher on the anxiety trait (7.49 vs. 4.90; Z = 7.284; *p* < 0.0001) and anxiety state (6.02 vs. 4.55; Z = 4.417; *p* < 0.0001) scales. Additionally, AUD group means were significantly higher for the NEO-FFI neuroticism (7.26 vs. 4.55; Z = 8.287; *p* < 0.0001) and openness (5.11 vs. 4.50; Z = 2.472; *p* = 0134) scales. On the other hand, significantly lower scores were obtained by the AUD subjects on the NEO-FFI extraversion (5.11 vs. 6.72; Z = −5.051; *p* < 0.0001), agreeability (3.84 vs. 5.49; Z = −5.380; *p* < 0.0001), and conscientiousness (4.97 vs. 6.88; Z = −5.844; *p* < 0.0001) scales. The means and standard deviations for the STAI trait and state scale and the NEO-FFI trait sten scores for the AUD and controls are presented in Table 3.

The performed factorial ANOVA of the AUD and control subjects, NEO-FFI, STAI, and *5HTR1A* rs6295, showed several interactions influencing the NEO-FFI openness to experience scale variance. We observed a significant effect of alcohol dependency or lack of it (F_1,207_ = 12.58; *p* = 0.0005; η^2^ = 0.057). The power observed for this factor was 94%, and approximately 6% was explained by alcohol dependency or lack thereof on the openness to experience trait variance. Additionally, we observed a statistically significant effect of the *5HTR1A* rs6295 genotype and alcohol dependency or lack thereof (F_2,207_ = 7.21; *p* = 0.0009; η^2^ = 0.065) on the openness to experience scale variance. The power observed for this factor was 93%, and approximately 6.5% was explained by the polymorphism of the *5HTR1A* rs6295 and AUD, or lack thereof, on the trait’s score variance. Moreover, there was a statistically significant effect of *5HT1A* rs6295 genotype interaction and alcohol dependency or lack of it on the openness scale (F_2,207_ = 6.67; *p* = 0.0016; η^2^ = 0.061; Table 4, Figure 1). The power observed for this factor was 91%, and approximately 6% was explained by the polymorphism of the *5HTR1A* rs6295 and alcohol dependency or lack thereof on openness trait score variance. The results of the factorial ANOVA of the State-Trait Anxiety Inventory, NEO Five-Factor Personality Inventory traits, *5HTR1A* rs6295, AUD, and controls are presented in Table 4.

Performed post hoc analysis revealed that AUD subjects with the *5HTR1A* rs6295 GC and GG genotypes obtained significantly lower results on the NEO-FFI openness scale than AUD subjects with the CC genotype. Additionally, the AUD subjects with the CC genotype obtained significantly higher results on the openness scale than the controls with the CC, CG, and GG genotypes. Table 5. presents the results of the post hoc test.

## 3. Discussion

Risk factors for alcohol dependence are complex and varied. Biological and psychosocial factors influence the risk of dependence in women. In our study, we included a biological factor, i.e., genetics, because although alcohol dependency is not directly heritable, certain polymorphic variants may predispose to substance dependency, and a psychosocial factor, i.e., personality and anxiety. We performed a case-control analysis involving 101 alcohol-dependent women. Our study focused on the serotonin 1A receptor single nucleotide polymorphism (rs6295), personality assessed by the NEO Five-Factor Inventory, and anxiety using the State Trait Anxiety Inventory. We also examined interactions between alcohol dependency, *5HTR1A* genotypes, personality traits, and anxiety indicators. To the author’s knowledge, this is the first published study presenting the association analysis in AUD females. Additionally, the available literature on substance-dependency phenotypes is not very rich. Hence, we will discuss the obtained results more from a neuropsychological perspective, presenting the plausible connections between substance dependency, personality traits, anxiety, and cognitive functions related to addiction-phenotype formation.

Numerous studies have shown that women tend to have a more active serotonin system compared to men [32]. This means that women typically have higher levels of tryptophan and a greater synthesis of 5-HT, both in the bloodstream and in the brain [33,34]. Animal studies have indicated that female animals have higher levels of serotonin in their blood and certain brain regions (brainstem, limbic forebrain, and hypothalamus/preoptic area) compared to males [35,36,37]. Studies on human cerebrospinal fluid have also suggested that women have a higher rate of serotonin metabolism in the brain [38,39]. Moreover, there is a fluctuation in serotonin levels throughout the 28-day menstrual cycle in women [40,41]. The results suggest that women may be more susceptible to serotonin alterations due to changes in the levels of its precursor. A clinical neuroimaging study using in vivo α-[11C]methyl-L-tryptophan positron emission tomography (PET) showed that women experienced a more significant effect of acute tryptophan depletion (ATD) on reduced brain 5-HT synthesis [42]. Additionally, other studies revealed that the impact of tryptophan depletion on verbal memory, mood, and emotion processing was more pronounced in women than in men [43,44,45,46].

As part of the first phase of the study, we performed a frequency analysis of genotypes and alleles in alcohol-dependent women and controls. The analysis did not show significant differences in the genotype and allele distribution of rs6295. Lee et al. [47] investigated the association between 5HT system genes and alcohol dependence. They studied 97 men with alcohol dependence and 76 men in a control group. For the 5-HT1A receptor, rs6295 genotype CC was significantly associated with alcohol dependence. A study by Wrzosek et al. [48] analysed serotonergic gene polymorphisms in a group of alcohol-dependent patients with and without suicidal behaviour, with negative results regarding rs6295. In a study by Stamatis et al. [49], the CC genotype modulated the influence of social anxiety on gambling attractiveness. The G allele of rs6265 may decrease 5HT1A auto receptor expression by influencing inhibitory transcription factor binding [23], which in turn may result in decreased serotonergic neurotransmission associated with inhibition [50] and performance monitoring [51], both important factors that may lead to the development of substance or behavioural addiction. Jacob et al. [52] in their study analysed personality traits in patients with attention-deficit/hyperactive disorder and personality disorders (PD). Concluding that the G allele of rs6295 may increase the risk of emotional-dramatic B cluster PD and decrease the risk of anxious-fearful C cluster personality disorders, pointing to crucial differences in emotional processing related to rs6265 variants.

Alcohol dependence is a complex disease in which genetics plays an important role, but not the only one [53]. The rs6295 polymorphism of the *5HTR1A* gene may influence the development of addiction [8]; however, many other factors also contribute to the disorder. Life environment, life experiences, and personality have a significant impact on the risk of developing addiction. The NEO-FFI questionnaire used in the study is a tool for assessing personality traits based on the Big Five model. It considers traits of personality such as extraversion, neuroticism, agreeableness, openness to experience, and conscientiousness [54]. Both genetics and environmental and personality factors are essential in understanding the mechanisms underlying alcohol dependence. Hence, the next step of our study was to analyse personality traits. We found that women with alcohol dependence obtained higher scores on the neuroticism and openness scales and lower values on the extraversion, agreeableness and conscientiousness scales. This suggests that certain personality traits are associated with a greater susceptibility to developing an addiction. We also performed a multivariate analysis, which showed that alcohol-dependent women who had the G/C and G/G variants of the rs6295 polymorphism in the *5HTR1A* gene had significantly lower openness scale scores than AUD women who had the CC genotype. Our results suggest that 5-HT1A transcriptional activity is associated with openness. These results confirm previous findings that the C allele of rs6295 is recognised as a binding site for the transcription factor Deaf1, which is only active in mature neuronal cells. The G allele turns off the binding of the transcriptional factors [23]. The mechanism by which Deaf1 exerts enhancer or repressor functions has yet to be fully understood. However, it is known that Deaf1 interacts with different transcriptional regulators and signalling proteins, such as glycogen synthase kinase 3β (GSK3β) [55], the transcription factor LMO4, the methyl-binding protein MeCP2, the E3 ubiquitin ligase Pellino1, and the DNA protein kinase subunit Ku70 [56]. Deaf1 has been linked to the differential regulation of pre- and post-synaptic 5HT1A receptor expression in mammals. Knockout of Deaf1 in mice led to an increase in 5HT1A autoreceptors but a decrease in post-synaptic 5-HT1A receptors, particularly in the prefrontal cortex (PFC) [57]. Similarly, in human heterozygosity, the G allele that inhibits Deaf1 binding was associated with reduced expression of 5-*HT1A* in the PFC compared to the C allele [58]. Interestingly, heterozygous depressed subjects showed reduced expression of PFC 5-HT1A receptors with the C allele, indicating a deactivation of Deaf1 enhancer activity [59].

Additionally, AUD subjects with the CC genotype scored lower on the openness scale than controls with the CC, CG, and GG variants. The other traits examined did not show significant associations. It is believed that AUD can stem from both excessive and deficient serotonin levels. There is evidence indicating the importance of categorising AUD patients as Type 1 or Type 2. Excessive serotonin is linked to Type 2, while deficient serotonin is associated with Type 1, which is more common in women [60]. Obtained in our research, a lower openness score in women with AUD compared to the control group may suggest overexpressed 5HT1A autoreceptors and diminished heteroreceptors in brain structures associated with this trait, such as the ventromedial prefrontal cortex (vmPFC), frontal and parietal structures in the default mode network (DMN), and dorsolateral prefrontal cortex (dlPFC) [61]. Increased inhibition by presynaptic autoreceptors could decrease overall serotonergic activity and contribute to AUD.

In addition, we analysed anxiety as a trait and as a state using the STAI questionnaire. Alcohol-dependent women showed significantly higher scores on both the STAI scales compared to the control group. This suggests that those with a tendency to experience higher levels of anxiety as a trait may be more prone to developing alcohol dependence, and alcohol-dependent individuals experience higher levels of anxiety in response to specific events or situations. Ellegaard et al. [62] found that personality traits such as low conscientiousness and high neuroticism were associated with poorer alcohol use disorder treatment outcomes. A study of 402 treatment-seeking clients found that a high score on the neuroticism scale was negatively associated with treatment completion. In addition, individuals with high neuroticism, openness, and extraversion scores or low conscientiousness scores were less likely to decrease their alcohol consumption to a reasonable level at follow-up after 6 months.

Beste et al. [51] suggest an impact of rs6265 on the reduction of neurophysiological processes underlying error monitoring and the attenuation of its behavioural consequences. Additionally, the study indicates that error-specific monitoring is associated with 5HRT1A-mediated neuronal processes. The ability to monitor and evaluate if an error has occurred is one of the cognitive functions crucial for survival in a changing environment [63]. On the neuropsychological level, those processes are assumed to be described by “error negativity” [64] or “error-related negativity” [65], which is associated with basal ganglia dopaminergic transmission and the anterior cingulate cortex [66,67]. Hence, “error-related negativity” is associated with dopaminergic dysfunctions [68,69,70,71,72]. According to Blum et al. [73], Reward Deficiency Syndrome present in addiction phenotypes, is associated with dopaminergic transmission, and error monitoring is part of cognitive dysfunction associated with substance dependency. Interestingly, the dopaminergic system in the prefrontal cortex is modulated by the serotonergic system [74].

In our study, we did not observe the interaction between rs6256 and anxiety measures. The effects of rs6256 on amygdala reactivity and anxiety were analysed by Fakra et al. [75]. In their study, the G allele (for both the GG and CG genotypes) was associated with a significant decrease in threat-related amygdala reactivity. Additionally, rs6256 and amygdala reactivity were associated with the anxiety trait. Interestingly, most studies associated the G allele with increased risk for anxiety [76], increased neuroticism, and harm avoidance [77]; all of those traits may be connected with increased reactivity of the amygdala [78,79,80,81]. In a study by Baas et al. [82], the *5HTR1A* rs6295 variants were associated with contextual fear independent of the anxiety trait, which confirms that failure to acquire cue contingencies impacts contextual fear-responding in association with trait anxiety. An association study of panic disorder (PD) in the Japanese population revealed a similar frequency of rs6265 variants in PD and controls, but in the subgroup of PD patients without agoraphobia, the GG genotype was significantly more frequent compared to PD patients with agoraphobia [83]. A study by Straube et al. [84] analysed PD patients with and without agoraphobia, revealing the association of the GG genotype with escapes during the behavioural avoidance test, measuring defence reactivity, and applied before cognitive-behavioural therapy (CBT). Additionally, the GG variant carriers presented a decrease in the effects of CBT-induced neural plasticity and normalisation of defence behaviour.

In our study, we analysed a group of females with AUD. Knowing that this particular area is highly underrepresented in terms of substance dependency research. The role of sex in the onset, progression, course and maintenance of alcohol dependency has been well documented [85]. A greater heritability of alcohol use disorders has been observed in males than in females [86,87]. In addition, men tend to exhibit higher alcohol consumption and a more prevalent lifetime history of alcohol dependence in comparison to women, with women being more likely to develop a greater number of negative physical and psychological outcomes related to alcohol use disorders (AUDs) than men [88]. Cloninger’s typology of alcoholism has demonstrated that there are distinct personality trait differences related to sex. Type II is predominantly observed in males [89]. Furthermore, women tend to score higher on the Reward Dependency and Harm Avoidance scale than men [90]; men tend to score higher on the Novelty Seeking scale [91]. Additionally, women obtain higher scores on the neuroticism scale, and men tend to obtain higher scores on the psychoticism scale [92]. A study has identified a correlation between high levels of neuroticism and anxiety in women and novelty-seeking and binge-drinking behaviours among university students [93]. The presented evidence indicates that certain traits may be associated with an increased risk for alcohol use disorder, with this association exhibiting a sex-dependent pattern. Further research should examine the influence of brain development and hormonal variations on the observed age-related differences. [85,94]. A significant number of studies failed to consider the potential influence of sex [88], on the results by employing exclusively male samples. Furthermore, when both sexes were included in the analysis, the female samples were often considerably smaller. Cultural-driven gender differences can evolve and change, whereas biological sex differences remain, though they are often overlooked [95].

The presented study is distinctive in its approach and methodology, offering a unique contribution to the field. We analysed mostly omitted in the addiction research study group comprised purely of women with alcohol use disorder. To the best of the authors’ knowledge, this is the first study conducted in a group of women, presenting comprehensive data regarding personality, anxiety, and, rarely analysed in terms of substance dependency, *5HTR1A* rs6265, which further enriched our study.

The presented study is subject to a number of limitations: (1) The study group consisted of 213 women (101 with alcohol use disorders and 112 control subjects). Although the present study provides valuable insights, the results would be enhanced if a larger cohort were included, thus increasing statistical power and enabling the drawing of firmer conclusions. (2) The study included Caucasian women with AUD. It is essential to exercise caution when extrapolating findings to broader populations, such as men and individuals from different ethnic backgrounds. To ensure the findings are representative of the wider population affected by alcohol use disorder, it would be beneficial to diversify the sample. (3) The assessment of personality traits and anxiety was conducted by administering self-report inventories. It is acknowledged that self-reported data may be susceptible to the effects of social desirability bias or subjective interpretation. (4) We analysed the personality-genotype interplay in AUD female subjects. Future studies should include more environmental factors related to alcohol use disorder susceptibility (e.g., early-life stressors, upbringing, history of traumatic events). (5) The study focused on the rs6265 polymorphic site within the *5HTR1A* gene. It is also possible that other *5HTR1A* genetic variants and epigenetic modifications may contribute to the propensity to develop AUD. An expansion of the genetic scope would facilitate a more comprehensive understanding. (6) The study included only women. Analysing male participants separately could yield insights into gender-specific vulnerabilities.

## 4. Materials and Methods

### 4.1. Participants

The study group comprised 213 female volunteers; 101 were diagnosed with alcohol use disorder (mean age = 45.74, SD = 11.11), and 112 were without any neuropsychiatric diagnoses (controls mean age = 45.32, SD = 10.19). The study was approved by the Bioethics Committee of the Medical District Council in Zielona Góra (KB-07/72/2017). All participants gave their written, informed consent. The study was conducted in the Independent Laboratory of Behavioural Genetics and Epigenetics. Participants with alcohol use disorder were recruited into a rehabilitation facility after at least three months of treatment and sobriety. Both AUD and control participants were examined using the Mini International Neuropsychiatric Interview (MINI) to confirm the AUD diagnoses and/or lack of any neuropsychiatric diseases in the case of the control group. Both groups filled the NEO Five-Factor Personality (NEO-FFI) and State-Trait Anxiety (STAI) Inventories.

### 4.2. Psychometric Measures

The Mini-International Neuropsychiatric Interview (MINI) is a structured diagnostic interview designed for the evaluation of diagnoses according to the criteria outlined in the Diagnostic and Statistical Manual of Mental Disorders, Fourth Edition (DSM-IV) and the International Classification of Diseases, Tenth Revision (ICD-10).

The NEO Five-Factor Inventory (NEO-FFI) comprises six components for the five traits analysed according to the FIVE-Factor Model (FFM). These are extroversion (assertiveness, positive emotions, warmth, sociability, emotion seeking, activity), conscientiousness (order, competence, striving for achievements, self-discipline, duty, consideration), openness to experience (values, feelings, fantasy, actions, aesthetics, ideas), neuroticism (hostility, susceptibility to stress depression, self-awareness, anxiety, impulsivity), and agreeableness (straightforwardness, trust, altruism, modesty, compliance, tenderness) [96].

The State-Trait Anxiety Inventory (STAI) is employed to assess anxiety. State anxiety may be described as a persistent predisposition towards the experience of stress, discomfort, and worry. Trait anxiety is characterised by fear and the temporary activation of the autonomic nervous system in response to specific situations [97].

The results of the NEO-FFI and STAI tests were converted to sten scores. In accordance with the Polish standards for adults [98,99], the raw scores were converted to sten scales. The resulting values corresponded to the following interpretations: 1–2 sten represented very low results, 3–4 sten indicated low results, 5–6 sten reflected average results, 7–8 sten indicated high results, and 9–10 sten corresponded to very high results.

### 4.3. Laboratory Procedures

Blood was collected from the study participants from the ulnar vein using standard Vacuette^®^ K3EDTA tubes (Greiner Bio-Ohne, Kremsmünster, Austria). Genomic DNA was purified from blood leukocytes using the QIAapm^®^ DNA Mini Kit (Qiagen, Hilden, Germany). Genotyping was conducted with the real-time PCR method using the LightSNiP probes (TIB MOLBIOL, Berlin, Germany) and the LightCycler^®^ FastStart DNA Master HybProbe (Roche Diagnostics, Penzberg, Germany) on the LightCycler480 II instrument (Roche Diagnostics, Penzberg, Germany). Fluorescence signals were plotted as a function of temperature to obtain melting curves for each sample. *5HTR1A* rs6295 variants were red at 52.34 °C for the G allele and at 60.82 °C for the C allele.

### 4.4. Statistical Analysis

A concordance between the genotype frequency distribution and Hardy-Weinberg equilibrium (HWE) was examined using the HWE software (https://wpcalc.com/en/equilibrium-hardy-weinberg/, online tool accessed on 5 April 2023).

The chi-square test was employed to conduct an association analysis between the *5HTR1A* rs6295 genotypes and alleles and AUD.

The condition of homogeneity of variance was met, as indicated by the Levene test (*p* > 0.05). The variables under analysis were not distributed normally. The STAI (trait, state) and NEO Five-Factor Inventory (neuroticism, extraversion, openness, agreeability, and conscientiousness) sten scores were compared between cases and controls using the U Mann–Whitney test.

The relationship between the *5HTR1A* rs6295 variants was investigated using a multivariate analysis of factor effects ANOVA. The study was conducted using genotyping data from the AUD and control subjects, the STAI trait and state scales, and the NEO Five-Factor Inventory traits. The analysis included the following variables: STAI/NEO-FFI scale, genetic feature, control and AUD (genetic feature × control and AUD).

All computations were performed using STATISTICA 13 (Tibco Software Inc, Palo Alto, CA, USA) for Windows (Microsoft Corporation, Redmond, WA, USA).

## 5. Conclusions

Our study presents a comprehensive set of data regarding alcohol use disorder in females, with significant differences in personality traits and anxiety measure scores between AUD and control subjects. We did not observe differences in genotype or allele frequencies for rs6265. Interestingly, we observed a significant effect of rs6295 genotype interaction and alcohol dependency, or lack thereof on the openness scale. In conclusion, this study presents a comprehensive overview of alcohol dependence in women. It offers valuable insights into this complex topic, contributing to a more nuanced understanding of substance use among this specific demographic. Additionally, these findings may have implications for developing prevention and intervention strategies tailored to individual genetic and, most importantly, personality and anxiety differences. They may also help identify individuals at higher risk of developing AUD, which is crucial for early diagnosis and treatment.

## Figures and Tables

**Figure 1 ijms-25-06563-f001:**
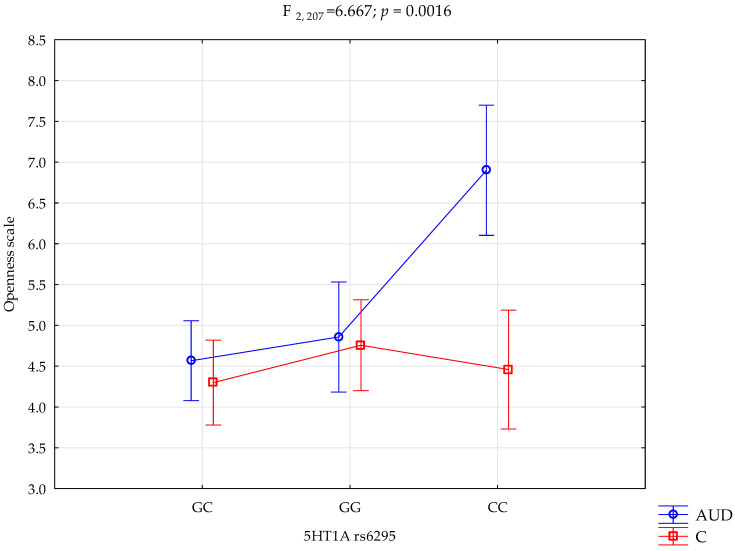
Interaction between the alcohol use disorder group (AUD), controls (C), *5HTR1A* rs6295, and the NEO-FFI openness scale.

**Table 1 ijms-25-06563-t001:** Hardy-Weinberg’s equilibrium of the *5HTR1A* gene rs6295 polymorphism for the AUD group and control subjects.

Hardy-Weinberg Equilibrium, Including Analysis for Ascertainment Bias		Observed (Expected)	Allele Freq	χ^2^ (*p*-Value)
AUD n = 101	G/C	53 (50.2)	p (G) = 0.54 q (C) = 0.46	0.318 (0.5727)
	G/G	28 (29.4)		
	C/C	20 (21.4)		
Control n = 112	G/C	47 (54.7)	p (G) = 0.58 q (C) = 0.42	2.224 (0.1359)
	G/G	41 (37.1)		
	C/C	24 (20.1)		

*p*-value, statistical significance, χ^2^ test; AUD, alcohol use disorder.

**Table 2 ijms-25-06563-t002:** Frequency of genotypes and alleles of *5HTR1A* gene rs6295 polymorphisms in the alcohol use disorder and control subjects.

	Genotypes			Alleles	
	G/C	G/G	C/C	G	C
	n (%)	n (%)	n (%)	n (%)	n (%)
AUD	53	28	20	109	93
n = 101	(52.48%)	(27.72%)	(19.80%)	(53.96%)	(46.04%)
Control	47	41	24	129	95
n = 112	(41.96%)	(36.61%)	(21.43%)	(57.59%)	(42.41%)
χ^2^	2.612			0.57	
(*p*-value)	(0.2709)			(0.4513)	

n, number of subjects, *p*-value, statistical significance, χ^2^ test, AUD, alcohol use disorder.

**Table 3 ijms-25-06563-t003:** STAI and NEO Five-Factor Inventory sten scores in the AUD group and controls.

NEO-FFI STAI	AUD (n = 101)	Control (n = 112)	Z	(*p*-Value)
STAI trait scale	7.49 ± 2.20	4.90 ± 2.30	7.284	0.0000 *
STAI state scale	6.02 ± 2.51	4.55 ± 2.15	4.417	0.0000 *
Neuroticism scale	7.26 ± 1.86	4.55 ± 2.09	8.287	0.0000 *
Extraversion scale	5.11 ± 2.24	6.72 ± 1.91	−5.051	0.0000 *
Openness scale	5.11 ± 2.14	4.50 ± 1.66	2.472	0.0134 *
Agreeability scale	3.84 ± 1.87	5.49 ± 2.19	−5.380	0.0000 *
Conscientiousness scale	4.97 ± 2.33	6.88 ± 2.00	−5.844	0.0000 *

*p*, statistical significance with Mann–Whitney U-test; *n*, number of subjects; M ± SD, mean ± standard deviation; *, statistically significant differences, AUD, alcohol use disorder.

**Table 4 ijms-25-06563-t004:** The 2 × 3 factorial ANOVA results for AUD and control subjects, NEO-FFI, STAI, and *5HTR1A* rs6295.

STAI NEO-FFI	Group	*5HTR1A* rs6295				ANOVA		
		G/C n = 100 M ± SD	G/G n = 69 M ± SD	C/C n = 44 M ± SD	Factor	F (*p*-Value)	η^2^	Power (Alpha = 0.05)
STAI trait scale	Alcohol use disorder (AUD); n = 101	7.41 ± 2.26	7.43 ± 2.15	7.80 ± 2.19	Intercept AUD/control 5HT1A rs6295 AUD/control × *5HTR1A* rs6295	F_1,207_ = 1405.78 (*p* < 0.0001) * F_1,207_ = 67.18 (*p* = 0.0001) * F_2,207_ = 0.08 (*p* = 0.9206) F_2,207_ = 0.77 (*p* = 0.4624)	0.871 0.245 0.0008 0.007	1.000 1.000 0.062 0.181
	Control; n = 112	4.91 ± 2.14	5.15 ± 2.67	4.46 ± 1.89				
STAI state scale	Alcohol use disorder (AUD); n = 101	6.07 ± 2.49	5.75 ± 2.49	6.25 ± 2.67	Intercept AUD/control 5HT1A rs6295 AUD/control × *5HTR1A* rs6295	F_1,207_ = 948.85 (*p* < 0.0001) * F_1,207_ = 19.64 (*p* < 0.0001) * F_2,207_ = 0.06 (*p* = 0.9454) F_2,207_ = 0.52 (*p* = 0.5923)	0.820 0.087 0.0005 0.005	1.000 0.993 0.058 0.136
	Control; n = 112	4.59 ± 2.23	4.68 ± 2.17	4.25 ± 2.01				
Neuroticism scale	Alcohol use disorder (AUD); n = 101	7.19 ± 1.81	7.29 ± 1.96	7.40 ± 1.93	Intercept AUD/control 5HT1A rs6295 AUD/control × *5HTR1A* rs6295	F_1,207_ = 1643.86 (*p* < 0.0001) * F_1,207_ = 91.23 (*p* < 0.0001) * F_2,207_ = 0.41 (*p* = 0.6627) F_2,207_ = 0.51 (*p* = 0.6009)	0.888 0.306 0.004 0.005	1.000 1.000 0.116 0.133
	Control; n = 112	4.45 ± 2.02	4.88 ± 2.24	4.21 ± 1.98				
Extraversion scale	Alcohol use disorder (AUD); n = 101	5.06 ± 2.21	5.25 ± 2.30	5.05 ± 2.35	Intercept AUD/control 5HT1A rs6295 AUD/control × *5HTR1A* rs6295	F_1,207_ = 1512.46 (*p* < 0.0001) * F_1,207_ = 29.10 (*p* < 0.0001) * F_2,207_ = 0.18 (*p* = 0.8348) F_2,207_ = 0.20 (*p* = 0.8175)	0.880 0.123 0.002 0.002	1.000 0.999 0.078 0.081
	Control; n = 112	6.60 ± 0.85	6.71 ± 1.94	7.00 ± 2.04				
Openness scale	Alcohol use disorder (AUD); n = 101	4.57 ± 1.95	4.86 ± 2.16	6.90 ± 1.68	Intercept AUD/control 5HT1A rs6295 AUD/control × *5HTR1A* rs6295	F_1,207_ = 1416.95 (*p* < 0.0001) * F_1,207_ = 12.58 (*p* = 0.0005) * F_2,207_ = 7.21 (*p* = 0.0009) * F_2,207_ = 6.67 (*p* = 0.0016) *	0.872 0.057 0.065 0.061	1.000 0.942 0.932 0.911
	Control; n = 112	4.30 ± 1.91	4.76 ± 1.43	4.46 ± 1.50				
Agreeability scale	Alcohol use disorder (AUD); n = 101	4.09 ± 1.83	3.57 ± 1.91	3.55 ± 1.90	Intercept AUD/control 5HT1A rs6295 AUD/control × *5HTR1A* rs6295	F_1,207_ = 979.81 (*p* < 0.0001) F_1,207_ = 38.03 (*p* < 0.0001) F_2,207_ = 1.06 (*p* = 0.3467) F_2,207_ = 1.22 (*p* = 0.2966)	0.826 0.155 0.010 0.012	1.000 0.999 0.235 0.265
	Control; n = 112	5.49 ± 2.13	5.15 ± 2.09	6.08 ± 2.39				
Conscientiousness scale	Alcohol use disorder (AUD); n = 101	5.02 ± 2.37	5.14 ± 2.40	4.60 ± 2.21	Intercept AUD/control 5HT1A rs6295 AUD/control × *5HTR1A* rs6295	F_1,207_ = 384.29 (*p* < 0.0001) F_1,207_ = 39.69 (*p* < 0.0001) F_2,207_ = 0.03 (*p* = 0.9744) F_2,207_ = 0.64 (*p* = 0.5282)	0.869 0.161 0.0003 0.006	1.000 0.999 0.054 0.156
	Control; n = 112	6.81 ± 2.03	6.80 ± 1.85	7.17 ± 2.26				

*, significant result; AUD, alcohol use disorder; M ± SD, mean ± standard deviation.

**Table 5 ijms-25-06563-t005:** Post hoc test (Least Significant Difference) of interactions between the alcohol use disorder group, controls, *5HTR1A* rs6295, and the NEO-FFI openness scale.

	{1}	{2}	{3}	{4}	{5}	{6}
	M = 4.57	M = 4.86	M = 6.90	M = 4.30	M = 4.76	M = 4.46
AUD G/C {1}		0.4918	0.0000 *	0.4603	0.6140	0.8091
AUD G/G {2}			0.0001 *	0.1968	0.8200	0.4290
AUD C/C {3}				0.0000 *	0.0000 *	0.0000 *
Control G/C {4}					0.2373	0.7240
Control G/G {5}						0.5227
Control C/C {6}						

*, significant statistical differences; M, mean.

## Data Availability

The genotyping and psychometric tests’ results are available upon request.

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
