# Peer review of "Association Study of Serotonin 1A Receptor Gene, Personality, and Anxiety in Women with Alcohol Use Disorder"

_ijms, 2024, doi:10.3390/ijms25126563_

Round 1
Reviewer 1 Report
Comments and Suggestions for Authors
The authors provide information about how the rs6295 polymorphism of the 5HTR1A gene in women with alcohol use disorder relates to a series of anxiety and other personality traits. No differences were observed in allele frequency in drinkers vs controls, suggesting no overall relationship between this site on the 5HT1R and drinking risk, even thought this SNP is related to other human neuropsychiatric conditions. Also, while drinkers had greater anxiety trait and state, and other personality measures, only the Openness scale was related both to alcohol dependence by genotype.
Thus, the negative overall findings provide new information about the likely complexity of the relationship between serotonin genotype, personality, and risk for heavy drinking.
The major question is whether AUDIT or other measures were taken. Alternately, AUD can be scored more as a continuum, rather than yes or no. Since the authors have the data, it would be useful to know if the level of AUD (or some other measure) related to personality and/or genotype.
The manuscript is overall good and clear, just two things to discuss more.
One is to provide greater insight into Openness as a variable, and what about this measure might explain why it relates to drinking while other personality traits do not. There is good discussion about what rs6265 might otherwise reflect, but the link to Openness is missing.
Is there an expected difference in men and women regarding serotonin? There is good discussion of sex differences with alcohol, missing just this one point.
Author Response
We sincerely thank you for all your great job and very valuable comments.
Below is the location of all changes, which can additionally be seen in the changes tracking panel.
- The major question is whether AUDIT or other measures were taken. Alternately, AUD can be scored more as a continuum, rather than yes or no. Since the authors have the data, it would be useful to know if the level of AUD (or some other measure) related to personality and/or genotype.
Thank you for this question. Unfortunately, recruiting females is more challenging due to their lower willingness to cooperate with researchers than men (clinical experience in Poland). Hence, the ability to gather data is decreased. It is regrettable that the AUDIT was not conducted, and all the data pertaining to both the patient and control groups are in the paper.
- One is to provide greater insight into Openness as a variable, and what about this measure might explain why it relates to drinking while other personality traits do not. There is good discussion about what rs6265 might otherwise reflect, but the link to Openness is missing.
Thank you for the suggestion. The information has been added in the Discussion section, lines 77-104.
- Is there an expected difference in men and women regarding serotonin? There is good discussion of sex differences with alcohol, missing just this one point.
Thank you for the suggestion. The information has been added in the Discussion section, lines 28-42.
- English needs some improvement.
Thank you for this comment; we have improved our English throughout the work.
Reviewer 2 Report
Comments and Suggestions for Authors
Dear authors, I have read the manuscript entitled "Case-control association analysis of serotonin 1A receptor gene, personality and anxiety in women with alcohol use disorder. The original work in the clinical area, touches on a rather delicate subject, present among the female population. The genetic investigations carried out by the authors of the manuscript provide valuable insights into this topic, contributing to a broader understanding of alcohol dependence in women.
The manuscript complies with the requirements of the journal, the results being presented through 5 tables and a figure. The large number of bibliographic references used for documentation are in agreement with the chosen topic. Although they were not mandatory, I appreciate that you also gave us some pertinent takeaways.
However, I have a few suggestions and questions:
1. The percentage of similarity (39%) is high. I ask the authors to reduce this percentage as much as possible.
2. The title is not very suggestive. Why case-control? You don't have a single case. It would be advisable for the authors to find another title.
3. You stated that the findings made in this study may have implications for the development of genetically tailored prevention and intervention strategies.
What kind of strategies did the authors think of?
4. English needs some improvement.
Comments on the Quality of English Language
English needs some improvement.
Author Response
We sincerely thank you for all your great job and very valuable comments.
Below is the location of all changes, which can additionally be seen in the changes tracking panel.
- The percentage of similarity (39%) is high. I ask the authors to reduce this percentage as much as possible.
Thank you for this suggestion. The similarity rate has been decreased.
- The title is not very suggestive. Why case-control? You don't have a single case. It would be advisable for the authors to find another title.
Thank you for this suggestion. The title was changed to: “Association study of serotonin 1A receptor gene, personality and anxiety in women with alcohol use disorder”
- You stated that the findings made in this study may have implications for the development of genetically tailored prevention and intervention strategies.
What kind of strategies did the authors think of?
Thank you for this question. Genetic testing can assist in identifying individuals at an increased risk of alcohol dependence. Based on the test results, prevention strategies and treatment can be adapted accordingly. Depending on the patient's genotype, the physician can select appropriate medications that will be more effective in reducing alcohol cravings or alleviating withdrawal symptoms. Individual personality traits can affect the effectiveness of therapy. Therefore, it is important to tailor the therapeutic approach to the individual patient. Support patients based on their individual needs and preferences. Education about genetic risk factors can facilitate understanding and acceptance.
- English needs some improvement.
Thank you for this comment; we have improved language, grammar and punctuation throughout the manuscript.
Round 2
Reviewer 1 Report
Comments and Suggestions for Authors
now acceptable
Comments on the Quality of English Languagenow good